# Stage-Specific Toxicity of Novaluron to Second-Instar *Spodoptera frugiperda* and *Plutella xylostella* and Associated Enzyme Responses

**DOI:** 10.3390/insects16101051

**Published:** 2025-10-15

**Authors:** Qing Feng, Jian Yang, Weikang Huang, Jingjing Jia, Jialing Wang, Fei Pan, Xuncong Ji

**Affiliations:** 1Institute of Plant Protection, Hainan Academy of Agricultural Sciences (Research Center of Quality Safety and Standards for Agro-Products, Hainan Academy of Agricultural Sciences), Haikou 571100, China; 2Hainan Key Laboratory for Control of Plant Diseases and Insect Pests, Haikou 571100, China; 3College of Tropical Crops, Yunnan Agricultural University, Puer 665099, China; 4General Station of Plant Protection in Hainan Province, Haikou 571100, China

**Keywords:** novaluron, *Spodoptera furgiperda*, *Plutella xylostella*, migratory pest, insecticidal activity, detoxifying enzymes, ecdysteroids

## Abstract

Investigations into the insecticidal activity of novaluron against the fall armyworm (*Spodoptera frugiperda*) and diamondback moth (*Plutella xylostella*) provide valuable data on its efficacy against these pests. Utilizing the leaf-dip method to assess toxicity to second-instar larvae and enzyme-linked immunosorbent assays (ELISA) to monitor changes in detoxifying enzymes and ecdysteroid levels, the study highlights distinct detoxification pathways in each pest species. The results indicate that physiological and ecological differences between the fall armyworm and diamondback moth influence their response to novaluron, offering valuable data for its application in integrated pest management.

## 1. Introduction

The fall armyworm, also known as the autumn cutworm, is a migratory pest native to the tropical and subtropical regions of the Americas [1,2,3]. Since its first detection in Puer City, Yunnan Province, China, in January 2019, where it was found to be infesting winter corn, the pest has been reported in different regions of China. By 9 July 2020, larvae of the fall armyworm had been identified in 1029 counties (districts) across 21 provinces (municipalities, regions), with an infested area of 65.53 × 10^4^ hectares [4]. The pest primarily affects sorghum, millet, wheat, buckwheat, peanuts, turmeric, banana, ginger, taro, rice, potato, rapeseed, and chili peppers, among other crops [5]. The larvae can damage crops throughout their entire growth period, feeding on leaves, stems, and even panicles. Moreover, they can cut off the roots of seedlings, leading to the death of crops in the seedling stage and ultimately causing severe economic losses [6]. Globally, the spread rate and damage range of the fall armyworm are also expanding [7]. Studies have indicated that the migratory capacity and adaptability of the fall armyworm enable it to survive and reproduce in diverse ecological environments, thereby exacerbating the severity of its damage [8]. The diamondback moth, a member of the family Plutellidae, is a worldwide migratory pest that is commonly found in vegetable-growing areas in China, primarily infesting cruciferous crops such as cabbage, radish, and cauliflower [9,10]. The larvae of the diamondback moth cause damage by feeding on leaves. Newly hatched larvae can bore into the tender leaves of vegetables to feed on the mesophyll; larvae at the end of the first instar and the beginning of the second instar emerge from the inside of the leaves to feed on the lower epidermis of the leaves, leaving the upper epidermis and forming white patches; third- and fourth-instar larvae can bite holes or notches in the leaves, and in severe cases, they can defoliate the entire leaf surface, leaving only the veins. This prevents normal photosynthesis in the plants, thereby affecting the growth and development of vegetables [11,12]. Due to the long-term use of chemical pesticides for control, the diamondback moth has developed resistance to various pesticides, including organochlorines, organophosphates, carbamates, pyrethroids, microbial insecticides (Bt), organotins, and macrolide insecticides [13,14]. Today, the diamondback moth has become one of the pests with the fastest development of resistance and the most severe pesticide resistance [12,15]. Currently, the efficacy of most conventional pesticides has significantly decreased, making it difficult to effectively control their damage. In the subtropical regions, the diamondback moth can occur and cause damage throughout the year, with more than 20 generations per year. In Hainan, it can have up to 22 generations in a year, with no overwintering or over-summering phenomena [16]. There is a clear overlap of generations, with large populations that can easily lead to outbreaks [17].

Recent studies have explored the toxic effects of novaluron on various pests. For instance, Thanasoponkul et al. [18] investigated the efficacy of novaluron against *Aedes aegypti* larvae, a major vector of mosquito-borne diseases. They found that novaluron at a concentration of 10 µg/L achieved similar mortality rates to 50 g/L of wet spent coffee grounds (wSCG) after 120 h of exposure. Additionally, sublethal combinations of wSCG and novaluron demonstrated a synergistic effect on larval mortality, suggesting a potential alternative control measure for mosquito larvae. These findings highlight the broad-spectrum insecticidal activity of novaluron and its potential application in integrated pest management strategies.

Despite these advancements, there are still significant gaps in our understanding of novaluron’s insecticidal activity. Specifically, there are no reports on the insecticidal activity of novaluron against the diamondback moth, and the research on the insecticidal activity of novaluron against the fall armyworm is still not comprehensive and in-depth [19,20]. Therefore, it is of great necessity to conduct research on the insecticidal activity of novaluron against these two pests.

In this study, the toxicity of novaluron to the second-instar larvae of these two pests was determined using the leaf-dip method. Corn leaves and cabbage treated with sublethal (LC_10_) and lethal concentrations (LC_50_) of novaluron insecticide were used to feed the second-instar larvae of the fall armyworm and diamondback moth, respectively. On this basis, the activity changes in detoxifying enzymes (CarE, P450, GST, and Ecd) and the target enzyme AChE in the larvae after feeding for 24 and 48 h were determined using the enzyme-linked immunosorbent assay (ELISA). This study aims to preliminarily explore the detoxification responses and physiological correlates of novaluron on the fall armyworm and diamondback moth, providing a theoretical basis for the effective control of these two pests in practical production. It is a beneficial supplement and improvement to the existing research and is expected to provide new ideas and methods for the control of the fall armyworm and diamondback moth in agricultural production, reducing crop losses caused by these two pests and ensuring the stable development of agricultural production.

## 2. Materials and Methods

### 2.1. Insects

The fall armyworm and diamondback moth were reared in the Integrated Insect Rearing Room of the Institute of Plant Protection, Hainan Academy of Agricultural Sciences (Hainan Tropical Agricultural Research Institute of Agricultural Product Quality and Safety and Standard Center). Rearing conditions were maintained at a temperature of 26 ± 1 °C, relative humidity of 70 ± 5%, and a photoperiod of L:D = 12 h:12 h. *S. frugiperda* larvae were fed on corn leaves grown without pesticide application, while *P. xylostella* larvae were fed on radish seedlings grown without pesticide application. After pupation, pupae were selected and placed in insect rearing cages. Emerging adults were provided with honey water for supplementary nutrition, and egg masses were collected for continuous rearing. Second-instar larvae with consistent growth and development were selected for the experiments.

### 2.2. Test Chemicals

The technical grade novaluron (≥98.5%) was purchased from Shanghai Aladdin Biochemical Technology Co., Ltd. (Shanghai, China). Reagent kits for CarE, cytochrome P450, GST, and AChE were obtained from Shanghai Kexing Trading Co., Ltd. (Shanghai, China). The insect Ecd ELISA detection kit was purchased from Shanghai Qiaodu Biotechnology Co., Ltd. (Shanghai, China).

### 2.3. Laboratory Bioassays of Toxicity

#### 2.3.1. Toxicity of Novaluron to Second-Instar *S. frugiperda*

The method of Xu et al. [21] was followed with slight modifications. The technical grade novaluron was first dissolved in DMF solvent and then diluted with a Tween-80 water solution to prepare five concentration gradients. For *S. frugiperda*, the concentrations were 0.01, 0.25, 0.5, 1, and 2 mg/L. For *P. xylostella*, the concentrations were 1, 2.5, 10, 25, and 50 mg/L. Two control groups were set up: one with the same proportion of DMF and Tween-80 as the highest concentration treatment group, and the other with distilled water as a blank control. The leaf-dip method was employed for the experiments. Corn leaves free of pesticides were cut into rectangular pieces (7 cm in length and 1.5 cm in width) and immersed in each concentration solution for 10 s, then air-dried. Ten dried corn leaves were placed in each Petri dish, and 10 second-instar *S. frugiperda* larvae starved for more than 4 h were introduced. Each treatment was replicated four times. Mortality was assessed by gently touching the larvae with a brush; larvae that showed no response, could not move normally, or whose body length was less than half of that of the control larvae were considered dead. Mortality was recorded after 48 h, and the mortality rate was calculated. The toxicity curves were plotted using Excel software 2023 (Microsoft Corporation, Redmond, WA, USA) to calculate the correlation coefficients. LC_10_, LC_50_, and the 95% confidence intervals were determined using SPSS 27.0 (IBM, Inc., Armonk, NY, USA).

#### 2.3.2. Toxicity of Novaluron to Second-Instar *P. xylostella*

The experimental method was the same as described in Section 2.3.1, but the test materials were different. In this experiment, pesticide-free cabbage leaves were used, cut into circular pieces with a diameter of 7 cm. The test subjects were second-instar *P. xylostella* larvae, with all other treatment conditions remaining unchanged.

### 2.4. Changes in Detoxifying Enzyme Activities and Ecd Levels

#### 2.4.1. Preparation of Test Solutions

Novaluron technical grade was formulated into test solutions at LC_10_ and LC_50_ concentrations using a Tween-80 water solution (Tween-80 was purchased from Shanghai Macklin Biochemical Co., Ltd., Shanghai, China) with a corresponding Tween-80 water solution as the solvent control (CK).

#### 2.4.2. Enzyme Source Preparation and Sample Processing

The specific steps were carried out according to the ELISA kit instructions (Shanghai Kexing Trading Co., Ltd.). Second-instar *S. frugiperda* larvae were fed on corn leaves treated with LC_10_ and LC_50_ concentrations of novaluron and the solvent control CK, while second-instar *P. xylostella* larvae were fed on cabbage leaves treated in the same manner. After 24 and 48 h of feeding, 10 live larvae were randomly selected from each treatment and placed in centrifuge tubes, which were then recorded and stored in liquid nitrogen at −20 °C for later use. The corresponding PBS buffer solution was added at a mass-to-volume ratio of 1:9.

#### 2.4.3. Determination of Detoxifying Enzyme Activities and Ecd Content

In this study, the enzyme-linked immunosorbent assay (ELISA) was employed to measure the relevant indicators. The activities of detoxifying enzymes, including carboxylesterase (CarE), cytochrome P450 (P450), and glutathione S-transferase (GST), were determined strictly according to the kit instructions. Additionally, the activity of acetylcholinesterase (AChE) was measured to assess neurophysiological function. The determination of ecdysteroid (Ecd) content was carried out following the instructions provided in the insect ecdysteroid ELISA detection kit. The specific experimental procedures and calculations are detailed in the Appendix A.

### 2.5. Data Analysis

Experimental data were used to construct toxicity curves and calculate correlation coefficients using Excel. LC_10_, LC_50_, and the 95% confidence intervals of novaluron for *S. frugiperda* and *P. xylostella* larvae were calculated using SPSS [22]. During the analysis of the effects of novaluron on enzyme activities and Ecd content in the larvae of the two species, the Grubbs’ G-value test was employed for data quality control to eliminate any potential outliers. Subsequently, one-way ANOVA followed by Tukey’s HSD test was conducted using SPSS to analyze the variance between different groups (with a significance level set at *p* = 0.05), and GraphPad Prism 10 was employed for data visualization.

## 3. Results

### 3.1. Toxicity of Novaluron to Second-Instar S. frugiperda and P. xylostella

The results of the laboratory bioassays of novaluron toxicity to second-instar larvae of *S. frugiperda* and *P. xylostella* are presented in Table 1. As shown in the table, novaluron exhibited higher toxicity to *S. frugiperda* larvae than to *P. xylostella* larvae. For more detailed information, the dose–response plots for both species are presented in Appendix A, and the model details for the dose–response analysis are provided in Appendix A. The raw mortality data by dose and replicate-level data for *S. frugiperda* are included in Appendix A, respectively. For *P. xylostella*, the corresponding data are included in Appendix A. These findings provide important references for selecting appropriate pesticides for the control of different pests.

### 3.2. Enzyme Activity Changes

#### 3.2.1. CarE Activity Assays

As shown in Figure 1A, the results of the effects of novaluron on CarE activity in *S. frugiperda* larvae indicated that after 24 h of treatment, there were no significant differences between the LC_10_ and LC_50_ treatment concentrations and the control group (*p* > 0.05). Similarly, after 48 h of treatment, no significant differences were observed between the LC_10_ and LC_50_ treatment concentrations and CK (*p* > 0.05). This suggests that under the experimental conditions, the metabolic response of *S. frugiperda* to novaluron in terms of CarE activity was not significant.

As shown in Figure 1B, the effects of novaluron on CarE activity in *P. xylostella* larvae exhibited a different trend. After 24 h of treatment, the CarE activity in both LC_10_ and LC_50_ treatment concentrations was significantly lower than that in CK (*p* < 0.05), indicating a significant inhibitory effect of novaluron on CarE activity in the early stage. However, after 48 h of treatment, the CarE activity in both LC_10_ and LC_50_ treatment concentrations was not significantly different from CK (*p* > 0.05), suggesting that *P. xylostella* could gradually restore CarE activity through its physiological regulatory mechanisms after a period of time.

#### 3.2.2. P450 Activity Assays

As shown in Figure 2A, the results of the effects of novaluron on P450 activity in *S. frugiperda* larvae indicated that after 24 h of treatment, the P450 activity in the LC_10_ treatment concentration was not significantly different from CK (*p* > 0.05), while the P450 activity in the LC_50_ treatment concentration was significantly higher than that in CK (*p* < 0.05). After 48 h of treatment, the P450 activity in the LC_10_ treatment concentration was significantly higher than that in CK (*p* < 0.01), and the P450 activity in the LC_50_ treatment concentration was also significantly higher than that in CK (*p* < 0.05). This suggests that in *S. frugiperda*, P450 activity significantly increased after 24 h of treatment with higher concentrations of novaluron and further enhanced after 48 h, indicating that P450 may play an important role in the detoxification metabolism of *S. frugiperda* against novaluron.

As shown in Figure 2B, the results of the effects of novaluron on P450 activity in *P. xylostella* larvae indicated that after 24 h of treatment, the P450 activity in both LC_10_ and LC_50_ treatment concentrations was not significantly different from CK (*p* > 0.05). However, after 48 h of treatment, the P450 activity in both LC_10_ and LC_50_ treatment concentrations was significantly higher than that in CK (*p* < 0.05). This suggests that *P. xylostella* exhibited a significant increase in P450 activity only after 48 h of treatment, indicating a relatively slower detoxification metabolic response to novaluron, but ultimately, it also upregulated P450 activity to cope with the pesticide stress.

#### 3.2.3. GST Activity Assays

As shown in Figure 3A, the results of the effects of novaluron on GST activity in *S. frugiperda* larvae indicated that after 24 h of treatment, the GST activity in both LC_10_ and LC_50_ treatment concentrations was not significantly different from CK (*p* > 0.05). After 48 h of treatment, the GST activity in the LC_10_ treatment concentration was significantly higher than that in CK (*p* < 0.01), and the GST activity in the LC_50_ treatment concentration was also significantly higher than that in CK (*p* < 0.05). This suggests that in *S. frugiperda*, GST activity significantly increased after longer-term treatment with novaluron, indicating that GST may play an important role in the detoxification metabolism of *S. frugiperda* against novaluron.

As shown in Figure 3B, the results of the effects of novaluron on GST activity in *P. xylostella* larvae indicated that after 24 h of treatment, the GST activity in both LC_10_ and LC_50_ treatment concentrations was not significantly different from CK (*p* > 0.05). After 48 h of treatment, the GST activity in the LC_10_ treatment concentration was significantly higher than that in CK (*p* < 0.01), and the GST activity in the LC_50_ treatment concentration was also significantly higher than that in CK (*p* < 0.05). This suggests that *P. xylostella* exhibited a significant increase in GST activity only after 48 h of treatment, indicating that it initiated GST participation in detoxification metabolism after longer-term treatment with novaluron.

#### 3.2.4. AChE Activity Assays

As shown in Figure 4A, the results of the effects of novaluron on AChE activity in *S. frugiperda* larvae indicated that after 24 h of treatment, AChE activity in both LC_10_ and LC_50_ treatment concentrations was not significantly different from that in CK (*p* > 0.05). However, after 48 h of treatment, AChE activity in the LC_10_ treatment concentration was significantly higher than that in CK (*p* < 0.05), and the AChE activity in the LC_50_ treatment concentration was extremely significantly higher than that in CK (*p* < 0.01). This suggests that after prolonged exposure to novaluron, AChE activity in *S. frugiperda* larvae significantly increased, indicating that AChE may play an important role in the detoxification metabolism of *S. frugiperda* against novaluron.

As shown in Figure 4B, the results of the effects of novaluron on AChE activity in *P. xylostella* larvae indicated that after 24 h of treatment, AChE activity in both LC_10_ and LC_50_ treatment concentrations was significantly higher than that in CK (*p* < 0.05). However, after 48 h of treatment, AChE activity in both LC_10_ and LC_50_ treatment concentrations was not significantly different from that in CK (*p* > 0.05). This suggests that *P. xylostella* exhibited a rapid AChE activity response to novaluron treatment in the early stage, but the AChE activity gradually returned to the control level over time, indicating that *P. xylostella* may adapt to novaluron stress through other mechanisms.

### 3.3. Ecd Content Determination

As illustrated in Figure 5A, the impact of novaluron on the Ecd content within *S. frugiperda* larvae was assessed. After 24 h of treatment, no significant differences in Ecd content were observed between the LC_10_ and LC_50_ treatment concentrations and CK (*p* > 0.05). Following 48 h of treatment, while the Ecd content in the LC_10_ treatment concentration remained non-significantly different from CK (*p* > 0.05), the Ecd content in the LC_50_ treatment concentration was significantly higher than that in CK (*p* < 0.01). This indicates that under higher concentrations of novaluron, the Ecd content in *S. frugiperda* larvae significantly increased after 48 h of exposure, suggesting that the synthesis of Ecd may be induced by the pesticide.

As depicted in Figure 5B, the effects of novaluron on the Ecd content within *P. xylostella* larvae were evaluated. After 24 h of treatment, no significant differences in Ecd content were detected between the LC_10_ and LC_50_ treatment concentrations and CK (*p* > 0.05). Similarly, after 48 h of treatment, no significant differences in Ecd content were observed between the LC_10_ and LC_50_ treatment concentrations and CK (*p* > 0.05). This suggests that under the experimental conditions, the Ecd content in *P. xylostella* larvae did not significantly change due to novaluron treatment. The lack of a significant response at the ecdysteroid level may be related to the metabolic action of detoxifying enzymes within the larvae.

## 4. Discussion

As global agricultural pests, *S. frugiperda* and *P. xylostella* cause significant economic losses to a variety of crops [23,24]. With the long-term use of traditional insecticides, these two pests have developed resistance to many commonly used pesticides, posing new challenges for pest control [25,26]. Therefore, the development of new, efficient, low-toxicity, and environmentally friendly insecticides is particularly important. This study focuses on novaluron, exploring its insecticidal activity and mechanisms of action against the larvae of *S. frugiperda* and *P. xylostella* through laboratory toxicity bioassays and analysis of detoxifying enzyme activities.

The results of this study show that the LC_50_ values of novaluron for second-instar *S. frugiperda* and *P. xylostella* were 0.480 mg/L and 3.479 mg/L, respectively, indicating higher insecticidal activity against *S. frugiperda* than *P. xylostella*. Compared to the patent research of Hailier Pharmaceutical Group Co., Ltd. (Chengyang East Industrial Park, Qingdao, China), the LC_50_ values for second-instar *S. frugiperda* and *P. xylostella* in this study were significantly lower than those for third-instar *P. xylostella*, beet armyworm, and cabbage worm mentioned in their patent [27]. This suggests that novaluron is more effective against younger larvae, likely due to their underdeveloped metabolic systems and increased sensitivity to novaluron. Additionally, its mode of action may be more effective against younger larvae. Therefore, novaluron has significant potential for the control of *S. frugiperda* and *P. xylostella* larvae.

In terms of detoxification mechanisms, this study found that *S. frugiperda* did not activate its detoxification mechanisms after 24 h of treatment. However, after 48 h, P450 and GST were involved in detoxification metabolism. Additionally, the activity of AChE was altered, which may be associated with the regulation of neurophysiological functions. In contrast, second-instar *P. xylostella* showed a significant decrease in AChE activity after 24 h of treatment, indicating a faster response to novaluron; after 48 h, P450 and GST were involved in detoxification metabolism, and the activities of other enzymes gradually recovered, which may be related to the physiological adaptation mechanisms of *P. xylostella*. These findings suggest that *P. xylostella* has a more rapid and robust detoxification response to novaluron compared to *S. frugiperda*. Additionally, the research by Zhu et al. [28] has shown that sublethal concentrations of insecticides significantly impact the activity of the detoxifying enzyme system in *P. xylostella*. This finding is akin to the effects of novaluron on P450 activity observed in the present study, thereby further substantiating the pivotal role of detoxifying enzymes in the development of insecticide resistance in insects. It has also been pointed out in other studies that *P. xylostella* exhibits a relatively slower detoxification metabolic response to insecticides, but ultimately copes with the insecticide stress by upregulating detoxifying enzyme activities [29]. This is in line with the results of the current study, which indicate that *P. xylostella* significantly upregulates P450 activity only after 48 h of novaluron treatment. However, the trend of CarE activity changes in *S. frugiperda* in this study differs from the results of Gao et al. [30]. In their study, CarE activity in *S. frugiperda* larvae showed no significant change when exposed to low concentrations of spinetoram (0.13 and 0.18 mg/L), but significantly increased at higher concentrations (0.40 and 0.57 mg/L). In contrast, in our study, we observed a significant increase in CarE activity even at lower concentrations of novaluron. This discrepancy may be related to the differences in the target of the pesticide action or species differences between *S. frugiperda* and *P. xylostella*. In contrast, in our study, we observed a significant increase in CarE activity even at lower concentrations of novaluron. This discrepancy may be related to the differences in the target of the pesticide action or species differences between *S. frugiperda* and *P. xylostella*. In addition, the study by Tian et al. [31] also shows that the changes in detoxifying enzyme activity of *P. xylostella* at different time points are more complex, which may be due to the unique mode of action of novaluron as a growth regulator. These results indicate that there are significant differences in detoxification mechanisms between *S. frugiperda* and *P. xylostella*: *S. frugiperda* only activates part of its detoxification mechanisms after prolonged exposure, while *P. xylostella* responds to novaluron at an early stage. These differences, which may be related to the physiological characteristics of the two pests and the mode of action of novaluron, provide important references for further research.

Furthermore, this study also explored the effects of novaluron on the Ecd content in *S. frugiperda* and *P. xylostella*. The results showed that after 24 h of treatment, the Ecd content in *S. frugiperda* did not change significantly due to novaluron treatment; however, after 48 h, the Ecd content in larvae treated with the LC_50_ concentration was significantly higher than that in the control group. This phenomenon may be related to the decrease in juvenile hormone (JH) levels. Previous studies have found that when JH concentration decreases, the synthesis of ecdysone 20E increases and activates downstream genes (such as *Broad-Complex*) through its receptor EcR/USP complex, thereby initiating the molting or metamorphosis program [32]. Therefore, the significant increase in Ecd content in *S. frugiperda* larvae after 48 h of treatment may be due to the decrease in JH levels. In contrast, novaluron had no significant effect on the Ecd content in second-instar *P. xylostella*. This may be related to the detoxification of novaluron by detoxifying enzymes in *P. xylostella*, resulting in no significant change in Ecd content due to novaluron treatment [33]. These results further highlight the significant differences in the response mechanisms of *S. frugiperda* and *P. xylostella* to novaluron: the ecdysteroid metabolism of *S. frugiperda* is significantly affected by novaluron, while *P. xylostella* copes with novaluron stress through other mechanisms (such as the regulation of detoxifying enzyme activity). While our study provides insights into the detoxification responses and physiological correlates of novaluron exposure, we acknowledge that a comprehensive understanding of insecticide detoxification mechanisms would benefit from the inclusion of additional metabolic pathways. These pathways include UGTs (glucuronidation), SULTs (sulfation), FMOs (flavin-containing monooxygenases), epoxide hydrolases (EHs), ALDH/ADH (aldehyde and alcohol dehydrogenases), antioxidant defenses (SOD, CAT, peroxidases), and transporters (ABC family). These pathways play crucial roles in the metabolism and detoxification of xenobiotics, including insecticides. For instance, studies have shown that UGTs significantly contribute to detoxification by facilitating the excretion of xenobiotics [34]. Additionally, antioxidant defenses, including SOD, CAT, and peroxidases, protect cells from oxidative stress induced by insecticide exposure. It has been reported that SOD and CAT activities are significantly increased in *Helicoverpa armigera* larvae exposed to neonicotinoid insecticides, indicating their role in mitigating oxidative damage [35]. Due to the scope and resource limitations of our study, we did not include these additional pathways in our experimental design. Future research should consider incorporating these pathways to provide a more complete picture of the insect’s response to novaluron and other insecticides.

In summary, this study reveals the differences in insecticidal activity and mechanisms of action of novaluron against the larvae of *S. frugiperda* and *P. xylostella* through laboratory toxicity bioassays, analysis of detoxifying enzyme activities, and ecdysteroid content. The results show that novaluron has satisfactory insecticidal activity against both *S. frugiperda* and *P. xylostella*, particularly against second-instar larvae. Significant differences in detoxification mechanisms and ecdysteroid metabolism were observed between the two species, which may be related to their physiological characteristics and the mode of action of novaluron. These findings provide a theoretical basis for the application of novaluron in pest control and offer important references for further research into its mechanisms of action and the development of more effective pest management strategies.

## 5. Conclusions

In this study, the insecticidal activity and detoxification responses and physiological correlates of novaluron on the larvae of *S. frugiperda* and *P. xylostella* were investigated through laboratory toxicity bioassays and analysis of detoxifying enzyme activities. The results demonstrated that novaluron exhibits satisfactory insecticidal activity against both *S. frugiperda* and *P. xylostella*, particularly against second instar larvae. Significant differences in detoxification responses and physiological correlates and ecdysteroid metabolism were observed between the two species: *S. frugiperda* only activated part of its detoxification responses after prolonged exposure, while *P. xylostella* responded to novaluron at an early stage. These differences are likely related to the physiological characteristics of the two pests and the mode of action of novaluron. This study provides a theoretical basis for the application of novaluron in pest control and offers important references for further research into its detoxification responses and physiological correlates and the development of more effective pest management strategies. Future research may focus on exploring the field efficacy of novaluron and its impact on non-target organisms, as well as further elucidating its detoxification responses and physiological correlates to develop more efficient integrated pest management strategies.

## Figures and Tables

**Figure 1 insects-16-01051-f001:**
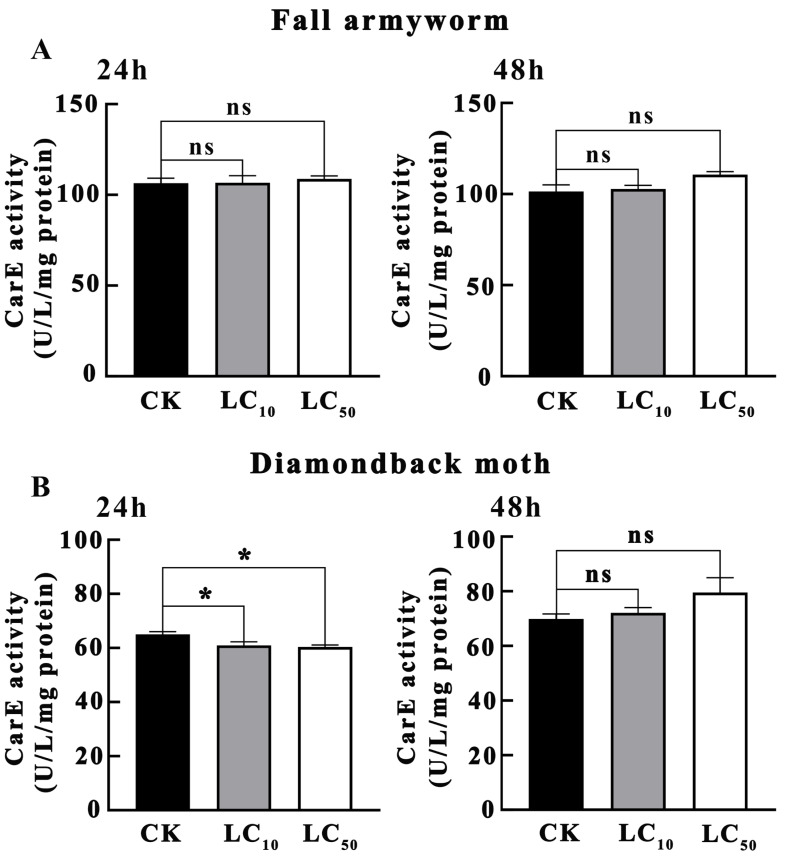
Effects of novaluron on CarE activity in larvae of fall armyworm and diamondback moth. (**A**) fall armyworm; (**B**) diamondback moth; ns: not significant; * *p* < 0.05; n = 4.

**Figure 2 insects-16-01051-f002:**
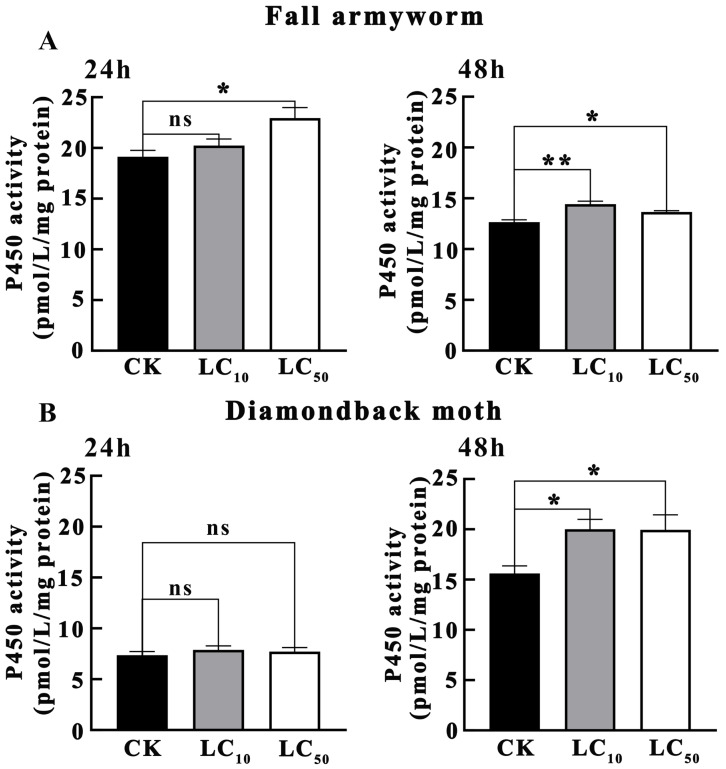
The effect of novaluron on the activity of P450 in the larvae of fall armyworm and diamondback moth. (**A**) fall armyworm; (**B**) diamondback moth; ns: not significant; * *p* < 0.05; ** *p* < 0.01; n = 4.

**Figure 3 insects-16-01051-f003:**
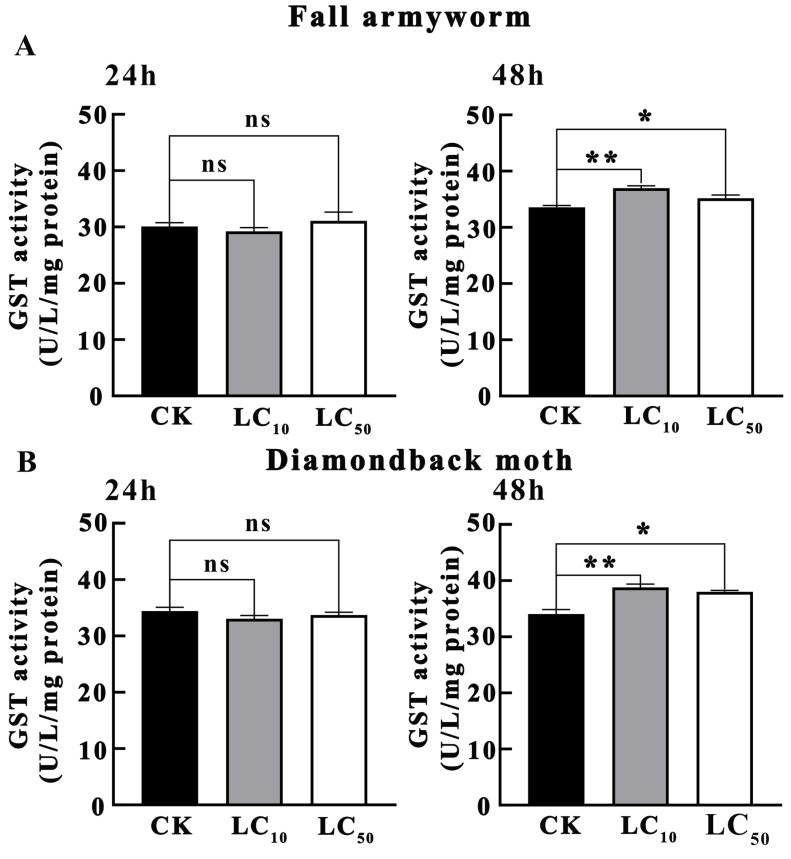
The effect of novaluron on the activity of GST in the larvae of fall armyworm and diamondback moth. (**A**) fall armyworm; (**B**) diamondback moth; ns: not significant; * *p* < 0.05; ** *p* < 0.01; n = 4.

**Figure 4 insects-16-01051-f004:**
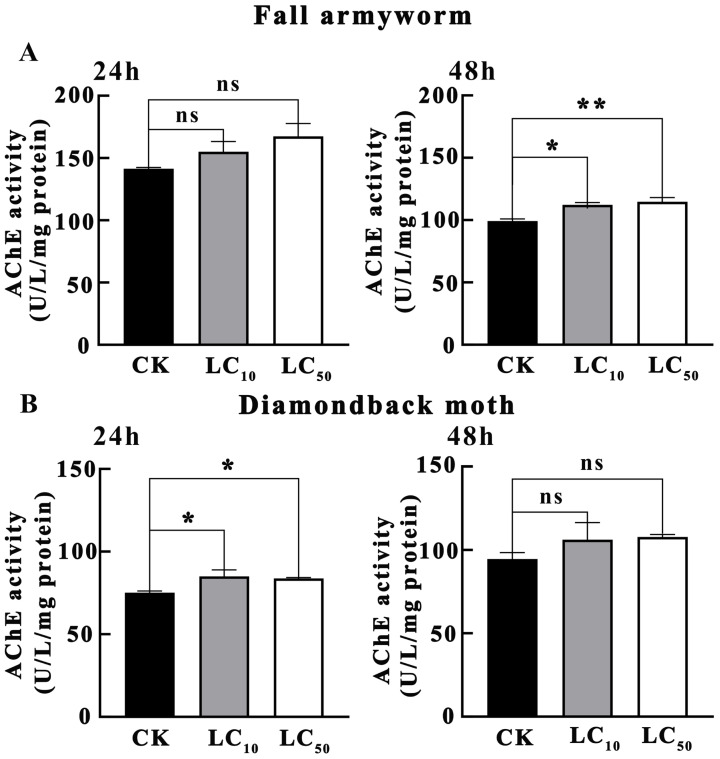
The effect of novaluron on the activity of AChE in the larvae of fall armyworm and diamondback moth. (**A**) fall armyworm; (**B**) diamondback moth; ns: not significant; * *p* < 0.05; ** *p* < 0.01; n = 4.

**Figure 5 insects-16-01051-f005:**
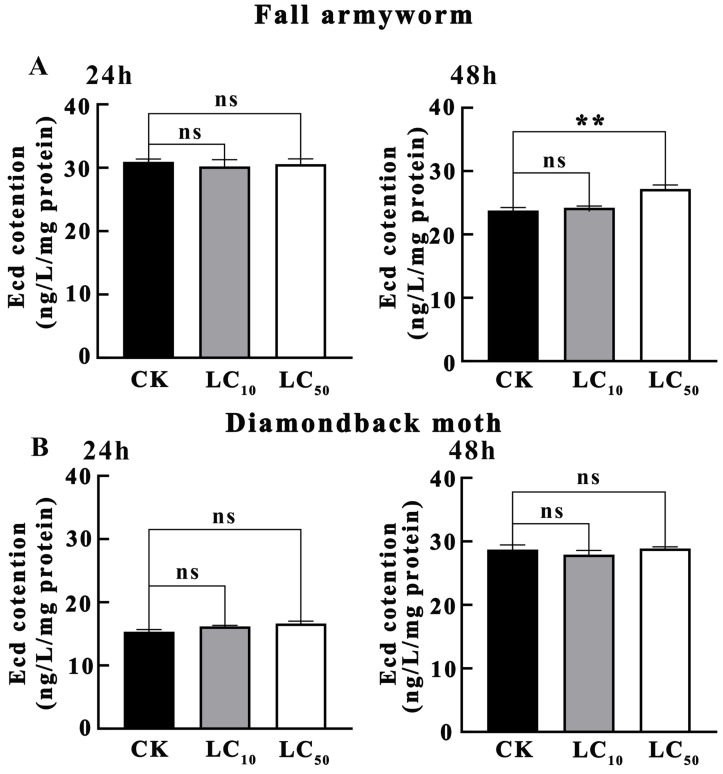
The effect of novaluron on the Ecd content in the larvae of fall armyworm and diamondback moth. (**A**) fall armyworm; (**B**) diamondback moth; ns: not significant; ** *p* < 0.01; n = 4.

**Table 1 insects-16-01051-t001:** Toxicity of novaluron to second-instar *S. frugiperda* and *P. xylostell*.

Target	Regression Equation	r	LC_10_ (95%FL) (mg/L)	LC_50_ (95%FL) (mg/L)	X^2^	df
*Spodoptera furgiperda*	y = 0.8353x + 0.2359	0.9737	0.087 (0.012–0.179)	0.480 (0.274–0.731)	33.509	18
*Plutella xylostella*	y = 0.005551x + 0.4326	0.9340	0.003 (0.000–0.080)	3.479 (0.308–9.472)	15.209	18

FL: Confidence limit.

## Data Availability

The original contributions presented in the study are included in the article/Appendix A, further inquiries can be directed to the corresponding authors.

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
