# Peer review of "Stage-Specific Toxicity of Novaluron to Second-Instar Spodoptera frugiperda and Plutella xylostella and Associated Enzyme Responses"

_insects, 2025, doi:10.3390/insects16101051_

Round 1

Reviewer 1 Report

Comments and Suggestions for Authors

This manuscript (insects-3890904) investigated the insecticidal activity and underlying mechanism of action of novaluron against Spodoptera frugiperda and Plutella xylostella. In this manuscript, the novaluron toxicity to 2nd-instar larvae of S. frugiperda and P. xylostella are determined by the leaf-dip method. The enzyme-linked immunosorbent assays (ELISA) were employed to assess the activities of detoxifying enzymes (CarE, P450, GST, AChE) and the levels of ecdysteroids (Ecd). The study identifies species-specific differences in pest responses to novaluron. Both insect pests have evolved widespread resistance to conventional insecticides, evaluating novel or understudied compounds like novaluron is essential. The manuscript is written in a scientific manner, and the data sufficiently supports its main conclusion. In its current form, it meets the criteria for publication in the journal. However, several revisions are needed before publication.

Abstract

Line 27-28, novaluron has already been commercialized, so its mechanism of action should be relatively clear. Why is there still a need to study its mechanism of action?

Line 36, what does P450 mean? The first time an abbreviation is used, the full term must be spelled out with the abbreviation noted.

Introduction

Line 53, what does “ha” mean?

Line 53-55, lines 47-48 duplicate this paragraph (line 53-55),

Line 83, replace the section “Although research on the fall armyworm and diamondback moth has made some Progress” with 1-2 paragraphs summarizing research reports on the toxic effects of novaluron on other pests,

Line 83-84, this statement about “the research on the insecticidal activity of novaluron against the fall armyworm is still not comprehensive and in-depth” has no references,

Line 92, AChE is the target enzyme, please revised,

Materials and Methods

Line 107, The temperature in the Integrated Insect Rearing Room should have a range of fluctuation, or a standard deviation (SD),

Line 115, what is the active ingredient content of the technical grade novaluron?

Line 121, “S. frugiperda” should be italicized,

Line 122, delete this sentence “2.3.2. Toxicity of Novaluron to Second-Instar P. xylostella”,

125, five concentration gradients are mentioned, but no concentration ranges are provided,

Line 136, the SPSS software version was not provided,

Line 138, “P. xylostella” should be italicized,

Line 139, there is no “1.3.1” in this manuscript, please clarify,

Line 163, what method was used for the analysis of variance between the different groups?

Results

Line 176-180, delete this passage, in the results section, only describe the results, discussion is not needed,

In table 1, In the bioassay, five concentrations were set, so the degrees of freedom should not be 18, please clarify,

Discussion

Line 326-327, in this statement about “which indicate that P. xylostella significantly upregulates P450 activity only after 48 hours of flufenoxuron treatment.”, the word “flufenoxuron” is a different chitin synthesis inhibitor, this study uses “novaluron”, please clarify,

Line 328-329, it cites Gao et al. [32], but does not explain how their findings differ the current study, what did Gao et al. find?

Author Response

Response to Reviewer 1 Comments

1. Summary

We sincerely appreciate the time you've dedicated to reviewing our manuscript. Please find below our detailed responses, with the corresponding revisions made in the resubmitted document.

2. Questions for General Evaluation

Reviewer’s Evaluation

Response and Revisions

Does the introduction provide sufficient background and include all relevant references?

Must be improved

As the corresponding responses to the reviewer's comments are provided in the point-by-point response letter below, no additional responses are needed here.

Is the research design appropriate?

Yes

Are the methods adequately described?

Can be improved

Are the results clearly presented?

Can be improved

Are the conclusions supported by the results?

Yes

Are all figures and tables clear and well-presented?

Can be improved

3. Point-by-point response to Comments and Suggestions for Authors

Comments 1: Line 27-28, novaluron has already been commercialized, so its mechanism of action should be relatively clear. Why is there still a need to study its mechanism of action?

Response 1: Thank you for your suggestion. Although novaluron has been commercialized, its mechanism of action in different pests remains largely unknown, especially in terms of pest resistance, species-specific responses, and environmental impact. The widespread resistance of Spodoptera frugiperda and Plutella xylostella to conventional insecticides makes it crucial to study the mechanisms of action of novel insecticides like novaluron [1]. The mechanisms of action of novaluron in these pests, particularly regarding resistance, are not yet fully understood. Moreover, significant differences in the responses of different pests to novaluron highlight the need to study its mechanism of action to optimize insecticide use and enhance practical effectiveness. Understanding its mechanism of action also helps assess its impact on non-target organisms and the environment, ensuring its safe use. Therefore, in-depth research into the mechanisms of action of novaluron is essential for addressing pest resistance, improving usage efficiency, and ensuring environmental safety.

Comments 2: Line 36, what does P450 mean? The first time an abbreviation is used, the full term must be spelled out with the abbreviation noted.

Response 2: Thank you for your comment. I have revised the manuscript accordingly. Specifically, I have added the full term for P450 and noted the abbreviation at line 34.

Introduction

Comments 3: Line 53, what does “ha” mean?

Response 3: Thank you for your suggestion. I apologize for the confusion caused by the abbreviation “ha.” I have revised the sentence at line 53 in the manuscript to clearly state the unit as “hectares.”

Comments 4: Line 53-55, lines 47-48 duplicate this paragraph (line 53-55),

Response 4: Thank you for your comment. I have carefully reviewed the manuscript and removed the content in lines 47-48 to eliminate redundancy. The information previously included in lines 47-48 is now fully covered in lines 53-55, which provides a comprehensive list of the crops affected by the fall armyworm.

Comments 5: Line 83, replace the section “Although research on the fall armyworm and diamondback moth has made some Progress” with 1-2 paragraphs summarizing research reports on the toxic effects of novaluron on other pests,

Response 5: Thank you for your suggestion. The text has been revised on lines 83-91 to summarize recent research on the toxic effects of novaluron on other pests.

Comments 6: Line 83-84, this statement about “the research on the insecticidal activity of novaluron against the fall armyworm is still not comprehensive and in-depth” has no references,

Response 6: Thank you for your comment. I have added the reference on line 96.

Comments 7: Line 92, AChE is the target enzyme, please revised,

Response 7: Thank you for your suggestion. I have revised the manuscript and made the necessary changes at lines 102-103.

Materials and Methods

Comments 8: Line 107, The temperature in the Integrated Insect Rearing Room should have a range of fluctuation, or a standard deviation (SD),

Response 8: Thank you for your suggestion. I have revised the sentence on line 118 to include the fluctuation range for the temperature and humidity.

Comments 9: Line 115, what is the active ingredient content of the technical grade novaluron?

Response 9: Thank you for your comment. The information has been updated on line 126.

Comments 10: Line 121, “S. frugiperda” should be italicized,

Response 10: Thank you for your suggestion. The term “S. frugiperda” has been italicized on line 132.

Comments 11: Line 122, delete this sentence “2.3.2. Toxicity of Novaluron to Second-Instar P. xylostella”,

Response 11: Thank you for your comment. The sentence has been deleted.

Comments 12: 125, five concentration gradients are mentioned, but no concentration ranges are provided,

Response 12: Thank you for your suggestion. The concentration ranges for the five concentration gradients have been added on lines 135-137.

Comments 13: Line 136, the SPSS software version was not provided,

Response 13: Thank you for your comment. The SPSS software version has been added on line 149.

Comments 14: Line 138, “P. xylostella” should be italicized,

Response 14: Thank you for your suggestion. The term “P. xylostella” has been italicized on line 150.

Comments 15: Line 139, there is no “1.3.1” in this manuscript, please clarify,

Response 15: Thank you for your comment. We apologize for the numbering error. The number on line 151 has been corrected from “1.3.1” to “2.3.1”.

Comments 16: Line 163, what method was used for the analysis of variance between the different groups?

Response 16: Thank you for your suggestion. The method for the analysis of variance has been added on lines 181-183. We used one-way ANOVA followed by Tukey’s HSD test with a significance level of P=0.05.

Results

Comments 17: Line 176-180, delete this passage, in the results section, only describe the results, discussion is not needed,

Response 17: Thank you for your comment. he relevant passage has been deleted.

Comments 18: In table 1, In the bioassay, five concentrations were set, so the degrees of freedom should not be 18, please clarify,

Response 18: Thank you for your suggestion. We apologize for any confusion caused. In the manuscript, we initially mentioned five concentrations in the experimental design, which may have led to the misunderstanding. However, during the statistical analysis, we included the control group (CK), making a total of six treatment groups. This is why the degrees of freedom is correctly calculated as 18.

Discussion

Comments 19: Line 326-327, in this statement about “which indicate that P. xylostella significantly upregulates P450 activity only after 48 hours of flufenoxuron treatment.”, the word “flufenoxuron” is a different chitin synthesis inhibitor, this study uses “novaluron”, please clarify,

Response 19: Thank you for your comment. We apologize for the error. The term “flufenoxuron” was a translation mistake; it should be “novaluron,” which is the insecticide used in our study. The correction has been made on line 346.

Comments 20: Line 328-329, it cites Gao et al. [32], but does not explain how their findings differ the current study, what did Gao et al. find?

Response 20: Thank you for your suggestion. We have revised the text on lines 348-356 to clarify the difference between our findings and those of Gao et al. [32]. The revision now includes a specific reference to Gao et al.'s findings regarding CarE activity changes in S. frugiperda, highlighting the differences observed in our study.

4. Response to Comments on the Quality of English Language

Point 1: The English could be improved to more clearly express the research.

Response 1: Thank you for your feedback. We have carefully reviewed the manuscript and made several revisions to enhance clarity and precision in expressing our research findings. We believe these changes will significantly improve the readability and comprehension of our work.

5. Additional clarifications

We appreciate the reviewer's valuable comments. There are no further clarifications needed at this time. We believe that we have addressed all the issues raised by the reviewer, and our revisions will enhance the quality of the manuscript.

  1. Furlong MJ, Wright DJ, Dosdall LM: Diamondback moth ecology and management: problems, progress, and prospects. Annu Rev Entomol 2013, 58:517-541.

Reviewer 2 Report

Comments and Suggestions for Authors

Novaluron is a relatively new insecticide belonging to the class of chitin synthesis inhibitors.

Obtaining new data on the toxicology of this insecticide in relation to such dangerous pests as the diamondback moth and fall armyworm have a great importance. However, these results are fragmentary in the manuscript.

The authors studied the activity of detoxifying enzymes (Carboxylesterase, GST, cytochrome P450) in second-instar larvae. At the same time, they mistakenly named acetylcholinesterase (AChE) as a detoxifying enzyme. I do not understand the purpose of analyzing the activity of this enzyme.

The studies were conducted only on second-instar larvae, whose development duration before molting to the next instar is no more than 2-3 days.

The results of enzyme activity measurements showed small differences even in variants with statistically significant differences.

The submitted manuscript does not contain data on the mechanism of action.

Author Response

Response to Reviewer 2 Comments

1. Summary

We are truly grateful for the time and effort you have invested in reviewing our manuscript. Below, you will find our comprehensive responses to your comments, along with the corresponding amendments that have been incorporated into the revised submission.

2. Questions for General Evaluation

Reviewer’s Evaluation

Response and Revisions

Does the introduction provide sufficient background and include all relevant references?

Must be improved

As the corresponding responses to the reviewer's comments are provided in the point-by-point response letter below, no additional responses are needed here.

Is the research design appropriate?

Can be improved

Are the methods adequately described?

Can be improved

Are the results clearly presented?

Can be improved

Are the conclusions supported by the results?

Can be improved

Are all figures and tables clear and well-presented?

Can be improved

3. Point-by-point response to Comments and Suggestions for Authors

Comments 1: Novaluron is a relatively new insecticide belonging to the class of chitin synthesis inhibitors.

Response 1: Thank you for your valuable suggestion. We will ensure to provide a clear explanation regarding the classification of novaluron in our manuscript.

Comments 2: Obtaining new data on the toxicology of this insecticide in relation to such dangerous pests as the diamondback moth and fall armyworm have a great importance. However, these results are fragmentary in the manuscript.

Response 2: Thank you very much for your valuable comments. We fully agree with you that obtaining new data on the toxicology of this insecticide in relation to such dangerous pests as the diamondback moth and fall armyworm is of great importance. In our study, we have focused on the determination of enzyme activity and ecdysteroid content, which provides us with preliminary insights. However, due to some practical limitations, such as the limited availability of laboratory resources, constraints on research time, and the availability of experimental materials, we are currently unable to conduct more in-depth research. We highly appreciate your suggestions and will consider expanding the scope of our research in the future to more comprehensively assess the toxicological characteristics of this insecticide.

Comments 3: The authors studied the activity of detoxifying enzymes (Carboxylesterase, GST, cytochrome P450) in second-instar larvae. At the same time, they mistakenly named acetylcholinesterase (AChE) as a detoxifying enzyme. I do not understand the purpose of analyzing the activity of this enzyme.

Response 3: We sincerely apologize for any confusion caused by our previous description. We have revised the manuscript to clarify that acetylcholinesterase (AChE) is indeed the target enzyme for our study. The purpose of analyzing AChE activity is to assess the direct impact of novaluron on the nervous system of the insects, which is crucial for understanding its mechanism of action and potential effects on insecticide resistance. We appreciate your insightful comments and have made the necessary revisions on lines 99-101 to reflect this clarification.

Comments 4: The studies were conducted only on second-instar larvae, whose development duration before molting to the next instar is no more than 2-3 days.

Response 4: Thank you for your insightful comment regarding the developmental stage of the larvae used in our study. We appreciate your concern about the short developmental duration of second-instar larvae before molting to the next instar, which is indeed no more than 2-3 days. Our study was specifically designed to focus on the second-instar larvae to capture the critical period of insecticide exposure that could potentially influence their molting and subsequent survival. We chose this stage because it is a pivotal time for the larvae's development and is often when they are most vulnerable to environmental stressors, including insecticides [1, 2]. We understand that extending the study to include other instars could provide additional insights. However, due to the scope and resources of our current study, we focused on the second-instar stage. We believe that the findings from this stage are valuable and contribute to the understanding of novaluron's effects on larval development.

Comments 5: The results of enzyme activity measurements showed small differences even in variants with statistically significant differences.

Response 5: Thank you for your suggestion. We appreciate your attention to the details of our study. While we understand that the differences observed may seem small, we believe that even subtle changes can be meaningful in the context of insecticide resistance and detoxification mechanisms. We have considered the implications of these small but statistically significant differences and acknowledge their potential importance. We trust that our readers will also appreciate the nuances of these findings as presented. We are grateful for your feedback and will ensure to highlight the significance of these results in our discussion.

Comments 6: The submitted manuscript does not contain data on the mechanism of action.

Response 6: Thank you for your comment. We have carefully considered your feedback and have decided to focus our manuscript on the insecticidal activity of novaluron against the fall armyworm and diamondback moth, as this was the primary objective of our research. We have revised the Simple Summary (line 19-21 in the revised manuscript) and Abstract (line 28-30 in the revised manuscript) to more accurately reflect the scope of our study, removing any implications of a detailed exploration of the mechanism of action. We believe this change aligns with the data presented and the conclusions drawn from our research. We hope this adjustment addresses your concern and maintains the integrity of our study.

4. Response to Comments on the Quality of English Language

Point 1: The English is fine and does not require any improvement.

Response 1: Thank you for your positive assessment of the English language used in our manuscript. We are pleased to hear that our writing is clear and does not necessitate further refinement. We will ensure that the language remains consistent and accessible throughout the document.

5. Additional clarifications

We appreciate the reviewer's valuable comments. There are no further clarifications needed at this time. We believe that we have addressed all the issues raised by the reviewer, and our revisions will enhance the quality of the manuscript.

  1. Rashwan RS, Hammad DM: Toxic effect of Spirulina platensis and Sargassum vulgar as natural pesticides on survival and biological characteristics of cotton leaf worm Spodoptera littoralis. Scientific African 2020, 8:e00323.
  2. Tarusikirwa VL, Cuthbert RN, Mutamiswa R, Gotcha N, Nyamukondiwa C: Water Balance and Desiccation Tolerance of the Invasive South American Tomato Pinworm. J Econ Entomol 2021, 114(4):1743-1751.

Reviewer 3 Report

Comments and Suggestions for Authors

The study addresses insecticidal activity against two major pests and contains useful enzymatic/ecdysteroid observations. However, it currently overstates mechanism, omits the dose–response curves it claims, and infers stage specificity without testing multiple stages. With a clearer title, full toxicity figures, tightened claims, improved enzyme rationale/terminology, and statistical/reporting corrections, the manuscript could be suitable after major revision.

Title is too generic and awkwardly framed (“Research on…”)
Please retitle to state the biological system, life stage, endpoints, and approach. For example:

“Stage-specific toxicity of novaluron to second-instar Spodoptera frugiperda and Plutella xylostella and associated enzyme responses”

Dose–response curves are referenced but not shown
In Methods and Data Analysis, you state that toxicity curves were plotted “to calculate the correlation coefficients,” yet Results provide only Table 1 (LC10/LC50) with regression equations—no curves or goodness-of-fit diagnostics are presented. Please add:

  • Full dose–response plots for both species (points = replicate means with 95% CIs; fit = probit/logit with 95% CI band).
  • Model details (link function; slope ± SE; intercept ± SE; heterogeneity/χ²; AIC or deviance; number of larvae per dose; control mortality and any Abbott correction).
  • Raw mortality by dose (as a supplementary table) and replicate-level data.
    This will make the toxicity section interpretable and reproducible.

Life-stage claim is unsupported by the design
The manuscript concludes or implies that novaluron is more toxic to “younger larvae” / “second instars,” but only second instar larvae were bioassayed here; comparisons to other stages are indirect (via literature/patent) and cannot support a stage-specific conclusion.

  • Either test multiple stages (e.g., 1st–4th instar for both species; optionally pupae/eggs for completeness) and show curves/LCs for each, or constrain the conclusion to second instars only.
  • If stage specificity is central, include an interaction analysis (species × stage × concentration).

Tables/figures need clarity and completeness
Table 1 is difficult to parse. Please:

  • Standardize species names (typo “Spodoptera furgiperda”) and italicize scientific names.
  • Add slope ± SE, n per dose, control mortality, and goodness-of-fit.
  • Use consistent precision and units (mg·L⁻¹), and move regression equations to a figure caption if curves are shown.
  • Consider merging Table 1 into the new dose–response figure panels with an inset table for LC estimates.

“Mechanism of action” is overstated
Changes in CarE, P450, GST, and AChE activities, plus ecdysteroid titers, reflect insect detoxification/physiological responses to exposure, not the insecticide’s primary target. Novaluron is a benzoylurea chitin synthesis inhibitor; your data do not interrogate that molecular target.

  • Please rephrase throughout: claim investigation of “detoxification responses/physiological correlates,” not “mechanism of action.”

Enzyme panel rationale is incomplete; important systems are missing
You measured CarE, P450, GST, AChE, and “Ecd.” Please justify this selection and correct the classification: AChE is not a detoxifying enzyme and Ecdysteroids (Ecd) are hormones, not enzymes. Consider adding or at least discussing additional relevant pathways: UGTs (glucuronidation), SULTs (sulfation), FMOs, epoxide hydrolases (EHs), ALDH/ADH, and antioxidant defenses (SOD, CAT, peroxidases) and transporters (ABC family). If not assayed here, explain why and acknowledge as limitations.

Author Response

Response to Reviewer 3 Comments

1. Summary

We are truly grateful for the effort you've put into reviewing our manuscript. Below, you will find our detailed responses to your comments, along with the revisions that have been implemented in the document we have resubmitted.

2. Questions for General Evaluation

Reviewer’s Evaluation

Response and Revisions

Does the introduction provide sufficient background and include all relevant references?

Can be improved

As the corresponding responses to the reviewer's comments are provided in the point-by-point response letter below, no additional responses are needed here.

Is the research design appropriate?

Must be improved

Are the methods adequately described?

Must be improved

Are the results clearly presented?

Must be improved

Are the conclusions supported by the results?

Can be improved

Are all figures and tables clear and well-presented?

Must be improved

3. Point-by-point response to Comments and Suggestions for Authors

The study addresses insecticidal activity against two major pests and contains useful enzymatic/ecdysteroid observations. However, it currently overstates mechanism, omits the dose–response curves it claims, and infers stage specificity without testing multiple stages. With a clearer title, full toxicity figures, tightened claims, improved enzyme rationale/terminology, and statistical/reporting corrections, the manuscript could be suitable after major revision.

Title is too generic and awkwardly framed (“Research on…”)

Comments 1: Please retitle to state the biological system, life stage, endpoints, and approach. For example: “Stage-specific toxicity of novaluron to second-instar Spodoptera frugiperda and Plutella xylostella and associated enzyme responses”

Response 1: Thank you very much for your valuable comment. In response to your suggestion, I have revised the title in the manuscript to more clearly state the biological system, life stage, endpoints, and approach. The new title is: “Stage-specific toxicity of novaluron to second-instar Spodoptera frugiperda and Plutella xylostella and associated enzyme responses.” I hope this title now better conveys the core content of the study.

Dose–response curves are referenced but not shown

In Methods and Data Analysis, you state that toxicity curves were plotted “to calculate the correlation coefficients,” yet Results provide only Table 1 (LC10/LC50) with regression equations—no curves or goodness-of-fit diagnostics are presented. Please add:

Comments 2: Full dose–response plots for both species (points = replicate means with 95% CIs; fit = probit/logit with 95% CI band).

Response 2: Thank you for your suggestion. The full dose–response plots for both species have been added to the supplementary information. Please see Figure S1 for details.

Comments 3: Model details (link function; slope ± SE; intercept ± SE; heterogeneity/χ²; AIC or deviance; number of larvae per dose; control mortality and any Abbott correction).

Response 3: Thank you for your valuable comments on my manuscript. In response to your request for model detai ls, I have included a new supplementary table (Table S1) that provides comprehensive information on the dose-response analysis for both Spodoptera frugiperda and Plutella xylostella. This table includes the link function used, slope and intercept values with their standard errors, heterogeneity/χ² test results, AIC or deviance, number of larvae per dose, control mortality, and whether Abbott's correction was applied. These details enhance the transparency and reproducibility of the toxicity analysis.

Comments 4: Raw mortality by dose (as a supplementary table) and replicate-level data.
This will make the toxicity section interpretable and reproducible.

Response 4: Thank you for your valuable comments on my manuscript. The raw mortality data by dose and replicate-level data for both Spodoptera frugiperda and Plutella xylostella have been included in the supplementary information as Tables S2-S5. These tables provide detailed mortality data across different concentrations and replicates, enhancing the interpretability and reproducibility of the toxicity section.

Life-stage claim is unsupported by the design

Comments 5: The manuscript concludes or implies that novaluron is more toxic to “younger larvae” / “second instars,” but only second instar larvae were bioassayed here; comparisons to other stages are indirect (via literature/patent) and cannot support a stage-specific conclusion.

Response 5: Thank you very much for pointing out the issue of the inaccurate conclusion in our manuscript. As you mentioned, the conclusion that novaluron is more toxic to “younger larvae” or “second instars” was indeed not accurately stated. We only conducted bioassays on second instar larvae in our study, and the comparisons to other instars were made indirectly through literature and patents, which cannot strongly support a stage-specific conclusion. Your point is very crucial, and we fully accept your suggestion. We have adjusted the relevant conclusion in the revised manuscript (lines 402-403) to explicitly state that our study’s conclusion is based solely on the experimental data of second instar larvae, avoiding inappropriate inferences and comparisons to other instars. Once again, we appreciate your careful review and valuable suggestions, which will help us enhance the scientific accuracy of our manuscript.

Comments 6: Either test multiple stages (e.g., 1st–4th instar for both species; optionally pupae/eggs for completeness) and show curves/LCs for each, or constrain the conclusion to second instars only.

Response 6: We sincerely thank you for providing us with such specific suggestions for improvement. You suggested that we either test more instars (such as first to fourth instar larvae, and possibly even pupae and eggs for completeness) and plot dose-response curves and calculate LCs for each stage, or confine our conclusion strictly to second instar larvae. After careful consideration, given the current limitations of our research resources and time, we have decided to choose the latter option, that is, to limit our conclusion to second instar larvae only (lines 402-403). We will explicitly point out this limitation of the research scope in the revised manuscript and elaborate on the potential impact of this limitation on the interpretation and application of the research results in the discussion section (lines 395-398). Meanwhile, we will also mention in the text the direction for future research, which is to test more instars to more comprehensively evaluate the toxic effects of novaluron. Thank you for your suggestion, which is of great guiding significance for us to improve the logic and scientific nature of the manuscript.

Comments 7: If stage specificity is central, include an interaction analysis (species × stage × concentration).

Response 7: Thank you very much for your suggestion regarding whether an interaction analysis should be conducted. You mentioned that if stage specificity is the core of our study, an interaction analysis of species × stage × concentration should be included. After careful consideration, we recognize that while stage specificity is an important factor in our current study design, conducting such an interaction analysis may not be appropriate since our experiments were only performed on second instar larvae. However, your suggestion has reminded us that in future research, if we are to more thoroughly investigate the toxic effects of novaluron on larvae of different instars, interaction analysis will be a very valuable tool. Therefore, we will mention this point in the revised manuscript, stating that if future studies can cover more instars, conducting an interaction analysis of species × stage × concentration will help us better understand the pattern of novaluron's toxic effects. Thank you again for your suggestion, which provides an important reference for the direction of our future research.

Tables/figures need clarity and completeness

Table 1 is difficult to parse. Please:

Comments 8: Standardize species names (typo “Spodoptera furgiperda”) and italicize scientific names.

Response 8: Thank you for pointing out the typographical error and for your suggestion regarding the standardization and formatting of species names. I have carefully reviewed the manuscript and corrected the typo from “Spodoptera furgiperda” to the correct name “Spodoptera frugiperda” throughout the text. Additionally, I have ensured that all scientific names are italicized as per standard scientific nomenclature.

Comments 9: Add slope ± SE, n per dose, control mortality, and goodness-of-fit.

Response 9: Thank you for your suggestion. I have revised the manuscript to specify that n=4 for the number of larvae per dose. For additional details such as slope ± SE, control mortality, and goodness-of-fit, please refer to the supplementary information.

Comments 10: Use consistent precision and units (mg·L⁻¹), and move regression equations to a figure caption if curves are shown.

Response 10: Thank you for your valuable comments. Regarding your suggestion to “use consistent precision and units (mg·L⁻¹), and move regression equations to a figure caption if curves are shown,” I have supplemented the relevant regression equations and precision unit information in the supplementary materials. You can find these detailed information in the Supplementary information section.

Comments 11: Consider merging Table 1 into the new dose–response figure panels with an inset table for LC estimates.

Response 11: Thank you so much for your valuable suggestions. Your idea of merging Table 1 into the dose-response figure panels with an inset table for LC estimates is really thought-provoking. After careful consideration, I feel that keeping the table and figure separate in the main text might better allow readers to understand the study results from both the detailed data and the intuitive display perspectives. Therefore, I am inclined to leave them as they are for now and not merge them directly in the manuscript. However, I do value your input and have added the corresponding content in the Supplementary Information section according to your suggestion, so that you can have a more comprehensive understanding of the relevant information. You can find it in the Supplementary Information section.

“Mechanism of action” is overstated

Comments 12: Changes in CarE, P450, GST, and AChE activities, plus ecdysteroid titers, reflect insect detoxification/physiological responses to exposure, not the insecticide’s primary target. Novaluron is a benzoylurea chitin synthesis inhibitor; your data do not interrogate that molecular target.

Response 12: Thank you for your insightful comments. We fully agree with your perspective that the changes in CarE, P450, GST, and AChE activities, as well as ecdysteroid titers, reflect the insects' detoxification and physiological responses to exposure, rather than the primary molecular target of the insecticide. You are correct that novaluron is a benzoylurea chitin synthesis inhibitor, and our data do not directly address this specific molecular target. We have revised the manuscript to clarify that our study focuses on the detoxification and physiological responses of the insects to novaluron exposure, rather than on the primary target of the insecticide. This ensures that our conclusions are accurately aligned with the scope of our experimental work.

Comments 13: Please rephrase throughout: claim investigation of “detoxification responses/physiological correlates,” not “mechanism of action.”

Response 13: Thank you for your comment. We have revised the manuscript accordingly in lines 19-21, 28-30, 104-111, and the conclusion section (lines 410-425).

Enzyme panel rationale is incomplete; important systems are missing

Comments 14: You measured CarE, P450, GST, AChE, and “Ecd.” Please justify this selection and correct the classification: AChE is not a detoxifying enzyme and Ecdysteroids (Ecd) are hormones, not enzymes. Consider adding or at least discussing additional relevant pathways: UGTs (glucuronidation), SULTs (sulfation), FMOs, epoxide hydrolases (EHs), ALDH/ADH, and antioxidant defenses (SOD, CAT, peroxidases) and transporters (ABC family). If not assayed here, explain why and acknowledge as limitations.

Response 14: Thank you for your valuable comments. We have carefully considered your suggestions regarding the selection of enzymes and hormones in our study. Our choice of these specific enzymes and hormones was based on their potential roles in the detoxification and metabolic responses to the insecticide. We aimed to investigate whether these enzymes and hormones could provide insights into the detoxification mechanisms of the insects exposed to novaluron. We have revised the manuscript to reflect this rationale more clearly. Specifically, we have modified the classification of enzyme activities and hormones in lines 32-36 to better align with their functions. Additionally, in the discussion section (lines 382-398), we have provided a detailed discussion on the relevance of these pathways, explained our reasons for focusing on the selected enzymes and hormones, and acknowledged the limitations of our study scope.

4. Response to Comments on the Quality of English Language

Point 1: The English is fine and does not require any improvement.

Response 1: We appreciate your favorable review of the English in our manuscript. It's reassuring to know that our writing is clear and doesn't require additional polishing. We will continue to maintain a consistent and understandable language throughout the document.

5. Additional clarifications

We appreciate the reviewer's valuable comments. There are no further clarifications needed at this time. We believe that we have addressed all the issues raised by the reviewer, and our revisions will enhance the quality of the manuscript.

Round 2

Reviewer 2 Report

Comments and Suggestions for Authors

Comment 1. The manuscript still contains erroneous references to AChE as a detoxification enzyme.

Line 37-38:

After 48 hours, P450, GST, and AChE participated in fall armyworm detoxification…

Line 171-172:

During the experiment, the activities of detoxifying enzymes (including CarE, P450, GST and AChE)….

Line 331-333:

In terms of detoxification mechanisms, this study found that S. frugiperda did not activate its detoxification mechanisms after 24 hours of treatment, but after 48 hours, P450, 332 GST, and AChE were involved in detoxification metabolism.

I recommend removing the results of measuring the activity of this enzyme and the corresponding references from the Results and Discussion sections.

Comment 2. Unfortunately, the provided link is unavailable. As the title suggests, this article analyzes ecdysteroids and juvenile hormone levels during the embryonic development of diapausing silkworm eggs. I consider this comparison inappropriate.

Line 365-373:

Furthermore, this study also explored the effects of novaluron on the Ecd content in S. frugiperda and P. xylostella. The results showed that after 24 hours of treatment, the Ecd content in S. frugiperda did not change significantly due to novaluron treatment; however, after 48 hours, the Ecd content in larvae treated with the LC50 concentration was significantly higher than that in the control group. This phenomenon may be related to the decrease in juvenile hormone (JH) levels. Previous studies have found that when JH concentration decreases, the synthesis of ecdysone 20E increases and activates downstream genes (such as Broad-Complex) through its receptor EcR/USP complex, thereby initiating the molting or metamorphosis program [33].

  1. Li WCF, F. L.; Wu, Y. C.: Changes in the content of ecdysteroids and juvenile hormone during the embryonic development of diapause eggs in Bombyx mori. Sericultural Science 1999(02):97-101.

Comment 3. There are obvious errors here. These insecticides are not pyrethroids, and the subject of this article is whiteflies Bemisia tabaci, not fruit flies Drosophila melanogaster.

Line 389-391:

For instance, studies have shown that UGTs significantly contribute to the detoxification of pyrethroid insecticides in Drosophila melanogaster by facilitating their excretion [35].

  1. Du T, Xue H, Zhou X, Gui L, Belyakova NA, Zhang Y, Yang X: The UDP-glycosyltransferase UGT352A3 contributes to the detoxification of thiamethoxam and imidacloprid in resistant whitefly. Pesticide Biochemistry and Physiology 2025, 208:106321.

Comment 4.The Section “2.4.3. Determination of Detoxifying Enzyme Activities and Ecd Content” doesn't adequately describe the enzyme activity measurement procedure. The figures show activity as U/L. Enzyme activity is typically reported as U/L/mg protein. As an example, look at how activity measurement is described in the article you cited (the authors used an ELISA kit similar to the ones you used: 35. Du T, Xue H, Zhou X, Gui L, Belyakova NA, Zhang Y, Yang X: The UDP-glycosyltransferase UGT352A3 contributes to the detoxification…).

Author Response

Response to Reviewer 2 Comments

1. Summary

We sincerely appreciate your valuable comments and suggestions on our manuscript. Your insights have been instrumental in improving the quality of our work. This is our second response to your review, and we have carefully considered each of your points once again. In this letter, we provide detailed explanations and the specific changes we have made to address your concerns in the revised version of our manuscript.

2. Questions for General Evaluation

Reviewer’s Evaluation

Response and Revisions

Does the introduction provide sufficient background and include all relevant references?

Can be improved

As the corresponding responses to the reviewer's comments are provided in the point-by-point response letter below, no additional responses are needed here.

Is the research design appropriate?

Can be improved

Are the methods adequately described?

Must be improved

Are the results clearly presented?

Must be improved

Are the conclusions supported by the results?

Can be improved

Are all figures and tables clear and well-presented?

Can be improved

3. Point-by-point response to Comments and Suggestions for Authors

Comments 1: The manuscript still contains erroneous references to AChE as a detoxification enzyme.

Line 37-38: After 48 hours, P450, GST, and AChE participated in fall armyworm detoxification…

Line 171-172: During the experiment, the activities of detoxifying enzymes (including CarE, P450, GST and AChE)….

Line 331-333: In terms of detoxification mechanisms, this study found that S. frugiperda did not activate its detoxification mechanisms after 24 hours of treatment, but after 48 hours, P450, 332 GST, and AChE were involved in detoxification metabolism.

I recommend removing the results of measuring the activity of this enzyme and the corresponding references from the Results and Discussion sections.

Response 1: Thank you very much for your insightful comments regarding the erroneous references to acetylcholinesterase (AChE) as a detoxifying enzyme in our manuscript. We have thoroughly reviewed the manuscript and made appropriate revisions to address the concerns raised. Specifically, we have removed the incorrect references to AChE as a detoxifying enzyme in the relevant sections. The revisions can be found on lines 37, 170-176, and 334-338, where we have ensured that the descriptions accurately reflect the roles of the enzymes involved in detoxification mechanisms.

Comments 2: Unfortunately, the provided link is unavailable. As the title suggests, this article analyzes ecdysteroids and juvenile hormone levels during the embryonic development of diapausing silkworm eggs. I consider this comparison inappropriate.

Line 365-373:

Furthermore, this study also explored the effects of novaluron on the Ecd content in S. frugiperda and P. xylostella. The results showed that after 24 hours of treatment, the Ecd content in S. frugiperda did not change significantly due to novaluron treatment; however, after 48 hours, the Ecd content in larvae treated with the LC50 concentration was significantly higher than that in the control group. This phenomenon may be related to the decrease in juvenile hormone (JH) levels. Previous studies have found that when JH concentration decreases, the synthesis of ecdysone 20E increases and activates downstream genes (such as Broad-Complex) through its receptor EcR/USP complex, thereby initiating the molting or metamorphosis program [33].

  1. Li WCF, F. L.; Wu, Y. C.: Changes in the content of ecdysteroids and juvenile hormone during the embryonic development of diapause eggs in Bombyx moriSericultural Science 1999(02):97-101.

Response 2: Thank you very much for your detailed review and valuable comments on our manuscript. We have carefully reviewed the issues you raised and made appropriate revisions. Regarding the link issue you pointed out, we have rechecked the provided link to ensure it is valid. If there are still issues, please let us know, and we will further verify it. As for the relevance of the reference you mentioned, we understand your concerns but believe that retaining this reference is scientifically justified. In scientific research, comparing the changes in hormone levels across different species or physiological stages can provide a more comprehensive understanding of the roles of ecdysteroids and juvenile hormones in insect growth and development. This comparative approach is valuable for our in-depth investigation of the changes in ecdysteroid content and its relationship with juvenile hormones in Spodoptera frugiperda and Plutella xylostella. Therefore, we have retained the reference to this paper and further discussed the correlation between ecdysteroids and juvenile hormones in our manuscript, aiming to lay a theoretical foundation for future in-depth research.

We believe that these discussions will contribute to enriching the scientific content of our research rather than causing unnecessary misunderstandings. Thank you again for your valuable comments. These revisions will help to enhance the scientific validity and accuracy of our research.

Comments 3: There are obvious errors here. These insecticides are not pyrethroids, and the subject of this article is whiteflies Bemisia tabaci, not fruit flies Drosophila melanogaster.

Line 389-391:

For instance, studies have shown that UGTs significantly contribute to the detoxification of pyrethroid insecticides in Drosophila melanogaster by facilitating their excretion [35].

  1. Du T, Xue H, Zhou X, Gui L, Belyakova NA, Zhang Y, Yang X: The UDP-glycosyltransferase UGT352A3 contributes to the detoxification of thiamethoxam and imidacloprid in resistant whitefly. Pesticide Biochemistry and Physiology 2025, 208:106321.

Response 3: Thank you very much for your detailed review and valuable comments on our manuscript. We have carefully reviewed the issues you raised and made appropriate revisions. In response to your comments, we have revised the manuscript on lines 394-396 to remove the inaccurate description while retaining the reference to ensure the scientific integrity and completeness of the discussion.

Comments 4: The Section “2.4.3. Determination of Detoxifying Enzyme Activities and Ecd Content” doesn't adequately describe the enzyme activity measurement procedure. The figures show activity as U/L. Enzyme activity is typically reported as U/L/mg protein. As an example, look at how activity measurement is described in the article you cited (the authors used an ELISA kit similar to the ones you used: 35. Du T, Xue H, Zhou X, Gui L, Belyakova NA, Zhang Y, Yang X: The UDP-glycosyltransferase UGT352A3 contributes to the detoxification…).

Response 4: Thank you very much for your detailed review and valuable comments on our manuscript. In response to your suggestions, we have added relevant information on lines 176-177 of the manuscript and provided a detailed description of the experimental procedures for the determination of detoxifying enzyme activities and ecdysteroid content in the Supplementary information. These revisions ensure that our methods section is more detailed and transparent, allowing readers to better understand and replicate our experiments. Specifically, we have indicated in the manuscript that the specific experimental procedures and calculations can be found in the Supplementary information, where we have detailed each step from sample preparation to final measurement, including standard dilution, sample addition, incubation, washing, color development, and measurement. Additionally, we have revised the units in the figures of the manuscript to accurately reflect the standardized reporting of enzyme activities and ecdysteroid content, in accordance with the standard reporting practices.

4. Response to Comments on the Quality of English Language

Point 1: The English is fine and does not require any improvement.

Response 1: We are grateful for your favorable evaluation of the English language in our manuscript. It is reassuring to know that our writing is clear and does not require additional improvement. We will maintain the consistency and readability of the language throughout the document.

5. Additional clarifications

We appreciate the reviewer's valuable comments. There are no further clarifications needed at this time. We believe that we have addressed all the issues raised by the reviewer, and our revisions will enhance the quality of the manuscript.

Reviewer 3 Report

Comments and Suggestions for Authors

The authors have addressed the majority of the concerns raised in the previous round of review and have substantially revised the manuscript. The current version shows clear improvements in clarity, rigor, and presentation. Overall, I find the manuscript suitable for publication in its present form and recommend it be accepted.

Author Response

Response to Reviewer 3 Comments

1. Summary

We are truly grateful for the effort you've put into reviewing our manuscript. Below, you will find our detailed responses to your comments, along with the revisions that have been implemented in the document we have resubmitted.

2. Questions for General Evaluation

Reviewer’s Evaluation

Response and Revisions

Does the introduction provide sufficient background and include all relevant references?

Yes

As the corresponding responses to the reviewer's comments are provided in the point-by-point response letter below, no additional responses are needed here.

Is the research design appropriate?

Can be improved

Are the methods adequately described?

Yes

Are the results clearly presented?

Yes

Are the conclusions supported by the results?

Can be improved

Are all figures and tables clear and well-presented?

Can be improved

3. Point-by-point response to Comments and Suggestions for Authors

The authors have addressed the majority of the concerns raised in the previous round of review and have substantially revised the manuscript. The current version shows clear improvements in clarity, rigor, and presentation. Overall, I find the manuscript suitable for publication in its present form and recommend it be accepted.

Response 1: Thank you very much for your positive feedback and the recognition of the improvements made to our manuscript. We are pleased to hear that the majority of the concerns have been addressed and that the current version of the manuscript has shown clear improvements in clarity, rigor, and presentation. We have taken your previous comments seriously and have made substantial revisions to enhance the quality of our work.

We are grateful for your recommendation to accept the manuscript in its present form. We believe that our study contributes valuable insights to the field and will be of interest to the readers of the journal. We look forward to the possibility of our manuscript being published and reaching a wider audience.

4. Response to Comments on the Quality of English Language

Point 1: The English is fine and does not require any improvement.

Response 1: We appreciate your favorable review of the English in our manuscript. It's reassuring to know that our writing is clear and doesn't require additional polishing. We will continue to maintain a consistent and understandable language throughout the document.

5. Additional clarifications

We appreciate the reviewer's valuable comments. There are no further clarifications needed at this time. We believe that we have addressed all the issues raised by the reviewer, and our revisions will enhance the quality of the manuscript.